# Diversity of Indicator and Dominant Plant Species along Elevation Gradients in Prince Mohammad Bin Salman Nature Reserve, KSA

Dhafer A. Al-Bakre 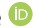

Biology Department, Faculty of Science, University of Tabuk, Tabuk 47512, Saudi Arabia; dalbakre@ut.edu.sa

**Abstract:** It is essential to protect and preserve biodiversity, especially in habitats in which natural resources are scarce. The differing flora and vegetation distribution on the broad, arid landscape at the Crown Prince Mohammed Bin Salman Nature Reserve has yet to be investigated. Based on GPS coordination and the transitional zone of plant communities, 48 symmetric plots of $50 \times 50$ m$^2$ were distributed evenly over six elevations. In this study, we recorded 70 species in 33 families and elucidated floristic traits correlated with elevation. High species richness was recorded for the families Fabaceae, Poaceae, Asteraceae, and Chenopodiaceae. High numbers of chamaephyte and phanerophyte species were observed. In the chorotype, the mono-regional component contained 47% of the species, the bi-regional component 35.7%, and multi-regional and worldwide species comprised 10% and 7%, respectively. This study noted the growth habits of 23 herbs, 15 shrubs, 10 trees, and a single species of grass, vine, climber, and mistletoe. Diversity indices, indicator species, dominant plant communities, and soil profiles were compared for the defined zones of elevation. Alpha and beta diversity were high at elevations of $\geq$1000, 800, and $\leq$100 ma.s.l., compared to elevations of 600 m, 400 m, and 200 m. The highest species richness and species turnover were recorded at elevations of $\geq$1000, 800, and $\leq$100 m, while species evenness was greater at elevations of 600, 400, and 200 m. Vegetation analyses and indicator species (based on relative abundance) showed species variation with elevation. Species domination was influenced by physical soil structure and soil chemistry. Microclimates, including temperature and relative humidity variations, were found to be a significant driver in the ecosystem, resulting in varying plant diversity and species distribution at different elevations. Through canonical correspondence analysis (CCA), we used an autocorrelation of elevations, plant species, and soil properties to identify three phytogeographic categories that were presumed to be a proxy of microclimate change: Category I: elevations 1000 m and 800 m, including *Retama raetam*, *Zilla Spinosa*, and *Vachellia gerrardii* linked with sandy soil; Category II: elevations 600 m and 400 m, including species *Haloxylon salicornicum, Rhazya stricta*, and *Leptadenia pyrotechnica* linked with enriched soils containing $CaCO_3$ and $HCO_3$ and having a clay texture; and Category III: elevations 200 m and 100 m, including *Zygophyllum coccineum*, *Tamarix nilotica*, and *Hyphaene thebaica*, which thrived in salinity and silt soils. The spatial vegetation patterns of the xeric environment and its transition zones in Prince Mohammed Bin Salman Nature Reserve were also documented. It is recommended that microclimate effects on species nominated for vegetation restoration or afforestation be considered for the optimal management of this important nature reserve.

**Keywords:** Prince Mohammad Bin Salman Nature Reserve; indicator species; species domination; arid landscape; plant diversity; elevation gradient

## 1. Introduction

Elevation represents a global environmental factor linked to decreasing atmospheric pressure and temperature, while abiotic factors such as sunlight and rainfall exhibit localized patterns [1]. Changes in non-living and living factors across elevation gradients can lead to different selection pressures, potentially resulting in local adaptation [2]. For many

years, ecologists have studied how species richness varies along elevational gradients, considering how factors such as climate, geographical area, and habitat diversity contribute to these patterns [3]. Research into large-scale ecological patterns has informed a general theory of species diversity and distributions, as well as an understanding of the various local and regional environmental factors that influence them [4,5].

Rapid climate effects coupled with anthropogenic activities diminish biodiversity values worldwide [6,7]. These direct and indirect factors lead to ongoing shifts in the status of at-risk species, and other species disappearing from their native habitats [8,9]. Spatial heterogeneity is an intrinsic indicator that affects species' incidence and abundance [10,11]; thus, it is essential to consider the environmental factors that impact species distribution as a precursor to assessing quantitative diversity [12].

At a wide spatial scale, elevation ranges above sea level are a critical abiotic factor, leading to variance in environmental gradients and the distribution of natural resources, such as air temperature, air humidity, water, and nutrient availability in soil, and in turn affecting the floristic diversity and spatial pattern of plant species [13]. Vegetation composition and plant diversity also vary within a limited habitat range [14]. The geographic site, slope, aspect orientation, and altitudinal gradients from the earth's ground level unevenly affect species distribution [15,16].

The distributional patterns of vegetation communities are also influenced by species' responses to habitat heterogeneity, such as adaptation to drought and saline stresses and outcompeting other species to colonize specific habitats [17–19]. In xeric environments, specific species can dominate a wide range of habitats, which often occurs on a broad regional scale [20]. The predominant species can also overlap with monospecific species that have distinctive microhabitats [21]. This makes the evaluation of species expansion to other vegetation zones or efforts to determine the spatial scale of a given species both costly and time-consuming [22].

The elevational gradient, with its diverse topographical features, such as mountains, serves as a refuge for species richness, providing protection against the prevalent arid conditions [23,24]. In the Kingdom of Saudi Arabia (KSA), biodiversity hotspots are frequently found in areas with significant elevational variations. The Sarawat Mountains in the southwestern region of KSA are a prime example, as they are home to an abundance of plant diversity. Numerous studies have highlighted this area as one of the most diverse habitats [25–27]. Another notable elevational range is Hijazi Mountain, which is located in the western region and is known for its rich flora [28]. However, the plant composition and vegetation structure across different elevational zones in the northwestern region of Saudi Arabia remain largely unexplored.

Prince Mohammad Bin Salman Nature Reserve (PMBSNR) is a new open nature reserve with public access roads comprising both agricultural and rural areas. It encompasses elevation variations involving diverse geological configurations, and environmental heterogeneity, providing a rich and biodiverse ecosystem. This study calculates the important value index to assess vegetation communities, whereas the incipient observations refer to the repetition of dominant and co-dominant species in most vegetation communities. This indicates an overlap of species distribution in the transitional zones at different elevational belts [29]. To overcome the species overlap that occurs when IVI is performed, it is worth adding further ecological predictors, such as indicator species (IS), which are often undertaken for monospecific species that colonize unique habitats [30]. A perfect indicator of a species is known to be exclusive to that habitat or group without incidence in other habitats [31,32]. Therefore, any species in a habitat with a recorded value greater than those of another habitat would be considered a "good indicator species" for that habitat [33]. Combined approaches IVI and IS are practical and robust tools that are suitable for this study.

The primary goal of this study is to investigate the floristic traits, diversity metrics, indicator species, and plant communities coupled with soil profiles along elevational gradients in PMBSNR. The assessment of elevational patterns in this study will provide

insight into the current state of vegetational distribution and help evaluate the loss of plant diversity, which can drastically alter ecosystems, subsequently threatening the existence of other species. This study also offers a baseline for the development of other programs, i.e., restoration and conservation management schemes, that require urgent action to conserve keystone species, focusing on rare, endangered life forms.

## 2. Methods

### 2.1. Study Area and Climate

Prince Mohammad Bin Salman Nature Reserve (PMBSNR) was established by royal order in June 2018 and has an area of 16,000 sq. km. This nature reserve is located between two giga projects—NEOM and the Red Sea Project—in northwest Saudi Arabia (Figure 1). The principal aim of Prince Mohammad bin Salman Nature Reserve is to conserve the natural environment of animals and plants and restore ecological balance. PMBSNR has distinctive topography, with variable microclimates. The comparison in the available data meteorology of 5 years (July 2018–December 2022) for Almuwaylih station at a coastline area, elevation ≤ 100 m above sea level (a.s.l.), and Shigry station at a height ≥ 1000 m (a.s.l.), showed that the annual average of precipitation did not differ significantly between these two elevations. However, there were significant differences in the annual averages of relative humidity and temperatures. The general pattern of Almuwaylih station at elevation ≤ 100 m (a.s.l.) indicated a high relative humidity and temperature compared to Shigry station at an elevation ≥ 1000 m (a.s.l.) (Figure 2).

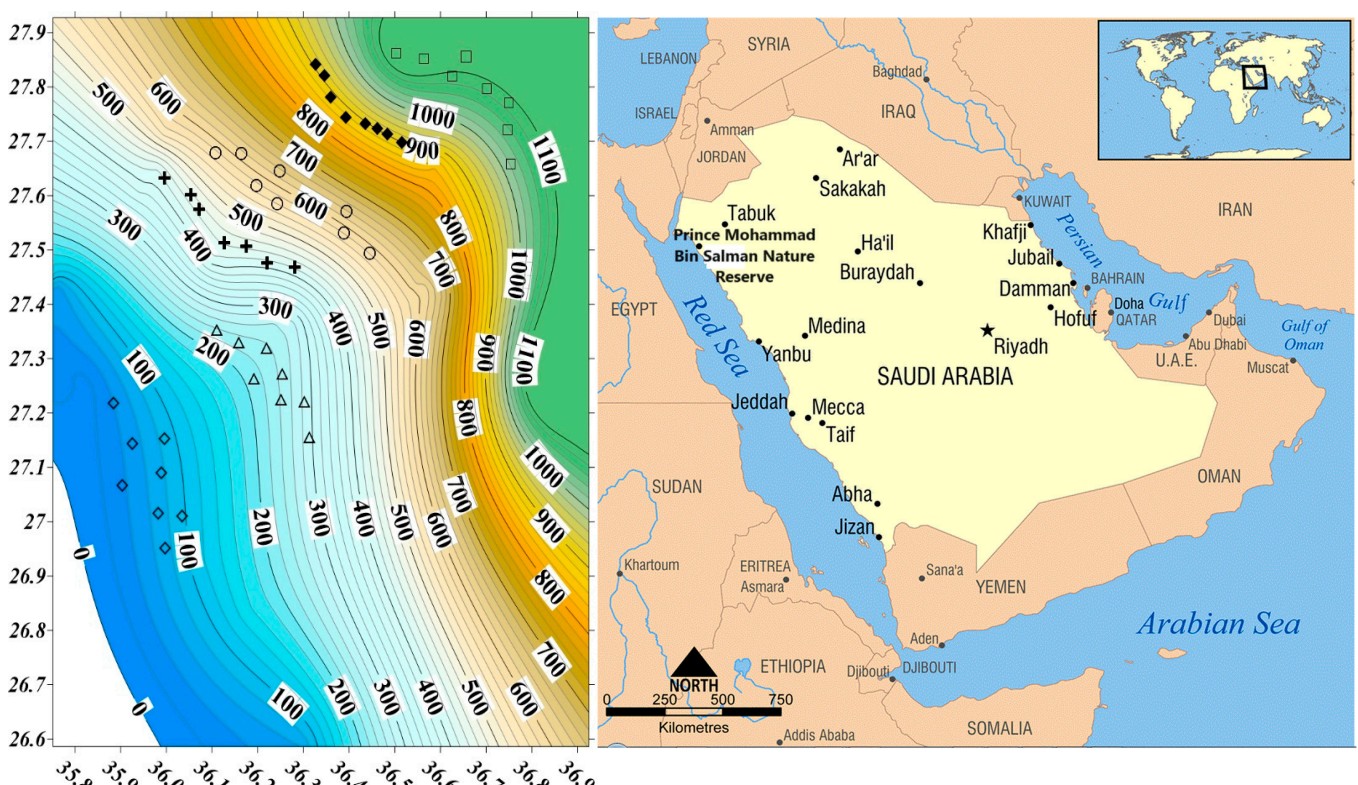

**Figure 1.** Saudi Arabia map (**right** photo) (https://www.worldmap1.com/saudi-arabia-map.asp), and (**left** photo) contour lines illustrate the elevational range between 0 and 1100 m above sea level (a.s.l.); the icon denotes the plots selected in each elevational belt: elevation ≥ 1000 m (□), elevation 800 m (■), elevation 600 m (○), elevation 400 m (+), elevation 200 m (∆), and elevation ≤ 100 m (◇).

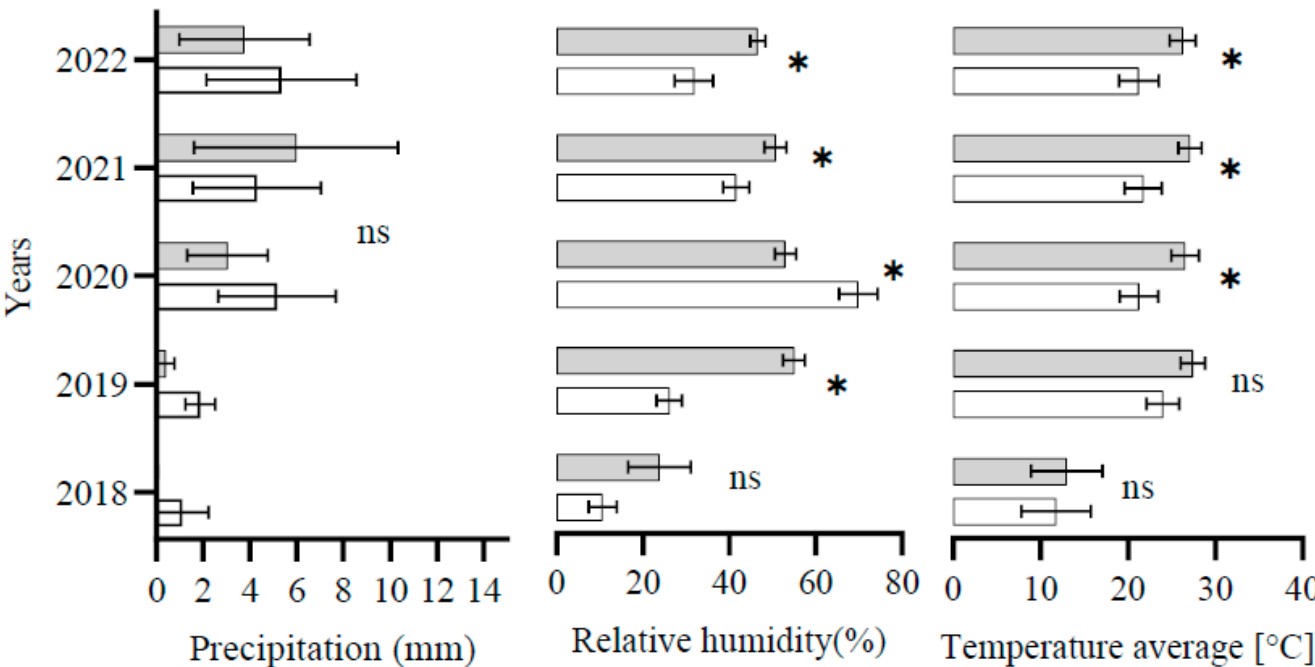

**Figure 2.** Multiple years of monthly average values of precipitation, relative humidity, and temperature from July 2018 to December 2022. Shed bar denotes data taken from Almuwaylih station at elevation $\leq$ 100 m a.s.l., and the open bar denotes data taken from Shigry station at elevation $\geq$ 1000 m a.s.l. (ns) denotes a non-significant *p*-value; an asterisk (*) denotes a significant *p*-value $\leq$ 0.05. Multiple *t*-test analysis, GraphPad Prism software version 9.5.1. (Data source, National Center Meteorology of KSA).

### 2.2. Field and Sampling Protocol

Fieldwork was based on changes in plant species and environmental factors throughout different zones, starting from higher peaks to sea level, whereas almost all topographical configurations exhibited a west-facing aspect. Plant species and soil profiles were sampled along elevational gradients from 1100 m to 10 m a.s.l. (Figure 1). GPS coordination defined the study sites at 200 m a.s.l. intervals of elevation gradients. A total of 48 sites were distributed among six elevations, in which each elevation encompassed eight representative plots of 50 × 50 m², as follows: elevation 1 ($\geq$1000 m) a.s.l., elevation 2 (800 m a.s.l.), elevation 3 (600 m a.s.l.), elevation 4 (400 m a.s.l.), elevation 5 (200 m a.s.l.), elevation 6 ($\leq$100 m a.s.l.).

### 2.3. Floristic and Vegetation Analysis

Soil sampling and analysis:

Soil sampling was conducted from the same quadrats, in which vegetative sampling was performed to correlate soil functional traits and species dominance. From each quadrat (n = 8) at each elevation, five random soil samples (0–40 cm) were collected in plastic bags and pooled as one composite sample. Initially, portions from each composite soil sample were separated and collected in duly labeled moisture tins to determine their soil moisture content (MC%) using the weight-loss method. All composite samples were duly labeled and transferred to the laboratory of the Biology Department, College of Science, University of Tabuk, Saudi Arabia, for further analysis. The samples were removed from plastic bags, spread over plastic sheets, air-dried at room temperature, and filtered through a 2-mm sieve to remove any debris. After that, the samples were stored in plastic bags for further analysis. Standard techniques were used for their physical and chemical examination [34]. The hydrometer method was used to examine the soil texture of the sand, silt, and clay fractions [35]. The oxidization method was used to measure the soil organic matter (OM%)

using $K_2Cr_2O_7$ [36]. Soil water extracts (1:5) were prepared to estimate soil electrical conductivity (EC) and pH [37]. A Calcimeter (Eijkelkamp, Agrisearch Equipment) was used to determine the calcium carbonate ($CaCO_3$) content. The titration method was used to examine the soluble anions (Cl and $SO_4$), while for soluble cations (Ca, Mg, Na, and K), a flame photometer was used [38].

*2.4. Data Analysis*

1—Floristic traits were identified based on the number of species, which were categorized by life form, growth form, chorotype, and the number of families along an elevational gradient. The nomenclature follows the KSA taxonomic database [39,40].

2—Diversity metrics were produced using (PAST—Paleontological Statistics software 4.03), which calculated alpha and beta diversity indices. The absence and presence of species was used to present the species richness Taxa-S, Shannon _H, Simpson_1-D, and Evenness_eˆH/S, and Beta-diversity Whittaker's index was used to measure species turnover [41,42].

3—This study presents a multivariate analysis-based Bray–Curtis index, hierarchy cluster analysis, and non-metric multidimensional scaling ordination (NMDS) using rank-based 48 plots via PAST software 4.03. Cluster analysis can be used to interpret the similarities between the plots. However, NMDS is more pronounced. Unlike other ordination techniques, NMDS is useful in cases of non-normality and discontinuous scales, and when some species are absent, resulting in data with zero values [43], followed by the indicator value of species (IS), which is a good tool for highlighting where the species' incidence typically occurs. The indicator value of species (IS) is calculated based on the relative abundance and frequency of the species, ranging from 0 (no indication) to 100 (a perfect indication) [44].

4—The important value index (IV) is based on the sum of relative density plus the relative covering of species. The IV determines the dominant, co-dominant, and other important species, as established by Bonham (2013) [45].

5—Canonical correspondence analysis (CCA) and positive/negative Pearson correlation (the heatmap) were applied to elucidate the relationship between the important value index (IV) and the measured physical and chemical soil variables along elevation variations.

## 3. Results

A floristic survey of the studied elevations highlighted 70 plant species belonging to 33 families (Table S1). Figure 3a shows that Fabaceae was prevalent, with nine species (12.85%), followed by Poaceae with seven species (10% each), and then five Asteraceae and Chenopodiaceae species (7% each). There were three species per family of Asclepiadaceae, Capparaceae, and Zygophyllaceae, comprising 4.28%. Approximately 9 families with two species each represented 2.85%, while the other 17 families were represented by 1.42% each, with one plant species. The chorological analysis is summarized in Figure 3b. Generally, mono-regional elements exhibited 47% (n = 33) of the identified species. Bi-regional elements represented 35.7% (n = 25), whereas pluri-regional and worldwide elements recorded 10%, and 7.1% (n = 7,5), respectively. This study classified the plant species into six life forms (Figure 3c): chamaephytes, which were a major life form, comprising 41.4% (n = 29), and Phanerophyte, which represented 32.85% (n = 23). Hemicryptophyte and Therophyte comprised 12.85% and 8.57% (n = 9,6), respectively, while other Geophytes, Cryptophytes, and Epiphytes represented 2.85%, 1.42%, and 1.42% (n = 2,1,1), respectively. The surveyed growth form species (Figure 3d) encompassed 23 herbs (32.85%), 16 shrubs (22.85%), 11 trees (15.71%), 10 shrublets (14.28%), 7 grasses (10%), and 1 species of vine, climber, and mistletoe (1.42%).

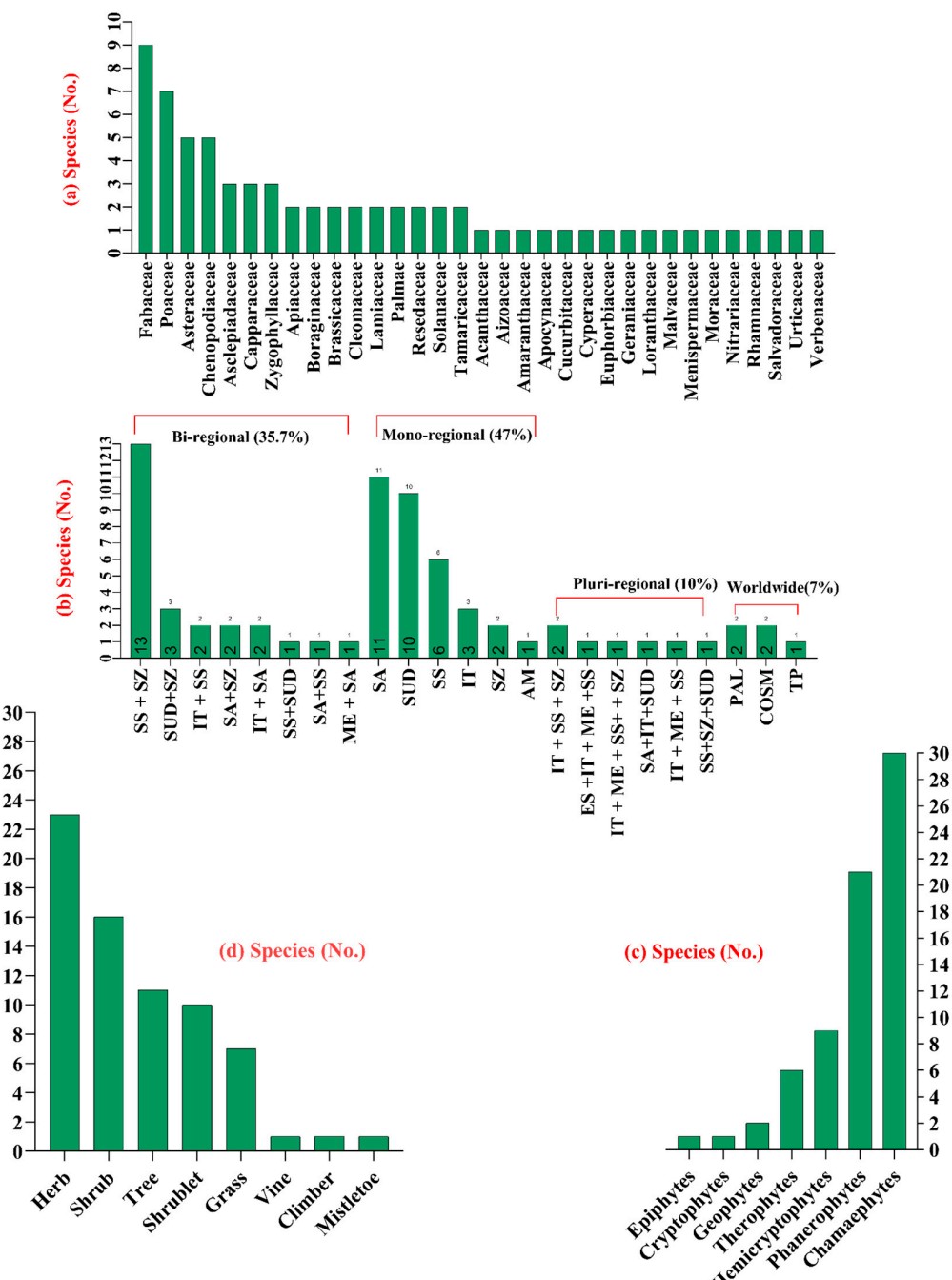

**Figure 3.** Floristic traits of the surveyed elevations. (**a**) Families, (**b**) chorotype spectra, (**c**) life forms, and (**d**) growth form. Chorotypes: American (AM); Cosmopolitan (COSM); Euro-Sibarian (ES); Irano-Turanian (IT); Mediterranean (ME); Paleotropical (PAL); Saharo-Sindian (SS); Sudano-Zambesian (SZ); Sudanian (SUD); Saharo-Arabian (SA); Tropical (TP).

Data for the diversity indices were distinguished along the studied elevations (Figure 4). The species richness of taxa was higher in elevations 1, 2, and 6 compared to elevations 3, 4, and 5. Elevation 3 had lower species richness compared to elevation 5, but it was not different from elevation 4 (Figure 4a). Elevation 6 had the highest Shannon index compared to the other elevations, while elevation 3 had the lowest Shannon index value (Figure 4b). Elevation 1 exhibited the lowest evenness compared to the other elevations, whereas elevation 5 represented the highest value of evenness (Figure 4c). Elevations 3 and 4 had a lower Simpson's index compared to elevations 1, 2, 5, and 6 (Figure 4d). In terms of

the Whittaker index, elevations 1 and 2 exhibited high values of species turnover compared to elevations 3, 4, 5, and 6. Elevation 6 had higher species turnover than elevations 4 and 5, while the means of species turnover were consistent for elevations 3, 4, and 5 (Figure 4e).

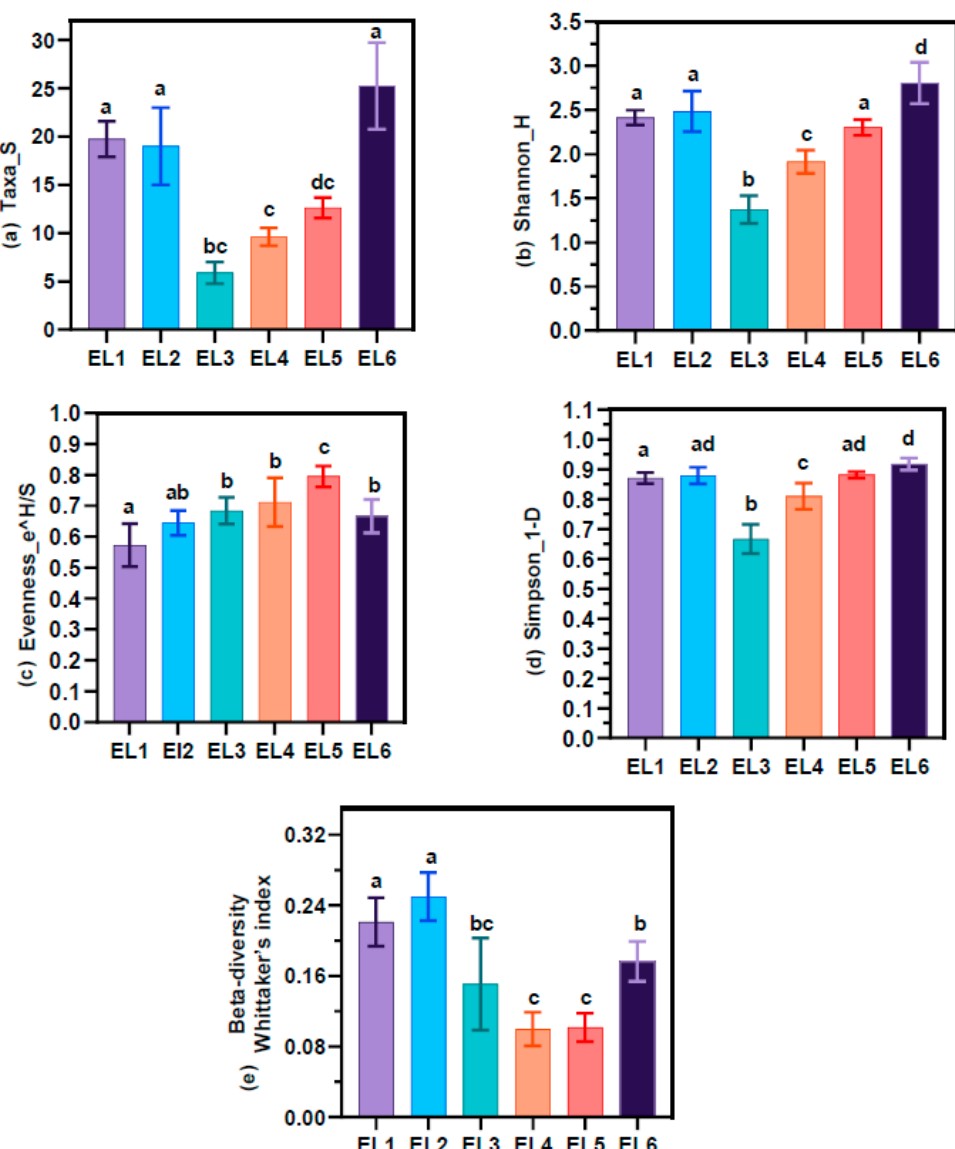

**Figure 4.** Plant diversity metrics of the surveyed elevations. (**a**) Taxa_ Species richness, (**b**) Shannon index, (**c**) Evenness_ eˆH/S, (**d**) Simpson_ 1-D, and (**e**) Beta diversity (Whittaker's index). Abbreviations EL1 = ≥1000 m (a.s.l.), EL2 = 800 m, EL3 = 600 m, EL4 = 400, EL5 = 200, and EL6 = ≤100 m. (EL) denotes elevation above sea level. PAST software (4.03) was used for the diversity indices, and GraphPad Prism 10.0.2 was used to create the graphs above. One-way ANOVA was performed using the Tukey test to compare means of elevations. The normality test was performed. Different letters denote statistical significance ($p < 0.05$).

Cluster analysis and NMDS data from all six elevations/48 plots showed a clear separation between elevations ≥ 1000 m (El1), 800 m (El2), 200 m (El5), and ≤100 m (El6); hence, significant changes in vegetation composition occurred. However, elevations of 600 m (El3) and 400 m (El4) were found to overlap with a close correlation, indicating a similarity in vegetation composition (Figure 5).

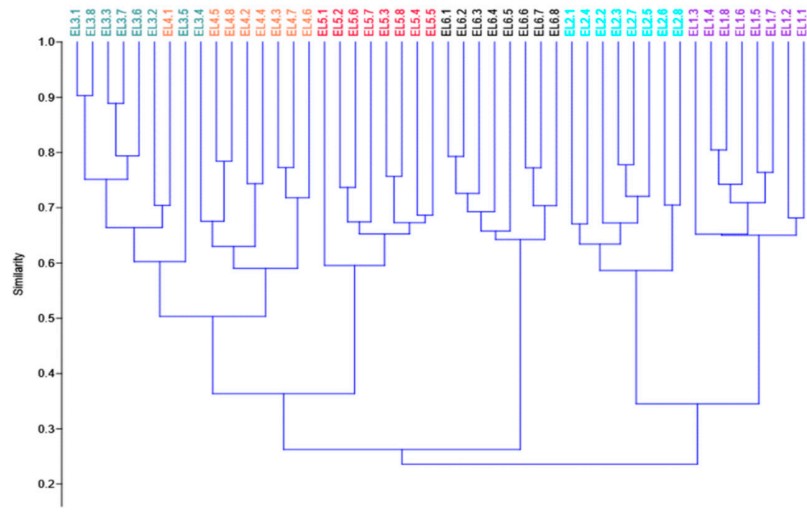

**Figure 5.** Bray–Curtis similarity index for the above cluster dendrogram and down NMDS, assembling 6 elevations/48 plots. El1.1 denotes elevation ≥ 1000 m/plot1, EL2.1 denotes elevation 800 m/plot1, EL3.1 denotes elevation 600 m/plot1, EL4.1 denotes elevation 400 m/plot1, EL5.1 denotes elevation 200 m/plot1, and EL6.1 denotes elevation ≤ 100 m/plot1. PAST software (4.03).

Sixty-six out of seventy species represented significant indicator values along elevational gradients ($p \leq 0.05$, Table 1). The most significant indicator species (IS) values were as follows: Elevation 1 recorded 20 indicator species representing 28 surveyed species in this elevational zone. The most prominent IS were *Retama raetam* and *Zilla spinosa* (IS = 1), which contributed 100%. It was determined that 18 out of 24 achieved significant indicator species for elevation 2, with the highest indicator value (IS = 0.956, 95.6%) for *Ziziphus spina-christi*. Elevation 6 included 23 indicator species out of 36 recorded species, as eight species belonging to this elevation attained the most significant indicator values (IS = 1, 100%): *Anastatica hierochuntica*, *Arthrocnemum macrostachyum*, *Cyperus bulbosus*, *Lasiurus scindicus*, *Nitraria retusa*, *Vachellia flava*, *Vachellia tortilis*, and *Zygophyllum coccineum*. Elevation 5 comprised 4 indicator species out of 13 plant species; the two most significant indicator species, *Senna holosericea* and *Tephrosia purpurea*, represented IS = 1100%. *Cleome droserifolia*, 1 species out of 12, showed a significant indicator value (IS = 0.5526, 55.26%) that occurred at elevation 4. No significant indicator species were identified from the seven plant species associated with elevation 3.

**Table 1.** The *p*-value of Indicator species (IS) orange colour and contribution percentage of species (%) yellow colour along elevational gradients, ELe1 ≥ 1000 asl, ELe2 = 800 m asl, ELe3 = 600 m asl, ELe4 = 400 m asl, ELe5 = 200 m asl, Ele6 ≤ 100 m asl. ANOSIM & SIMPER test—PAST package. Values are based on relative abundance and frequency.

| Indicator Species (IS) | ELe1 | ELe2 | ELe3 | ELe4 | ELe5 | ELe6 | ELe1 % | ELe2 % | ELe3 % | ELe4 % | ELe5 % | ELe6 % |
|---|---|---|---|---|---|---|---|---|---|---|---|---|
| *Abutilon bidentatum A. Rich.* | 1 | 0.1623 | 1 | 1 | 1 | 1 | 0 | 12.5 | 0 | 0 | 0 | 0 |
| *Achillea fragrantissima (Forssk.) Sch.Bip.* | 0.0006 | 1 | 1 | 1 | 1 | 1 | 50 | 0 | 0 | 0 | 0 | 0 |
| *Aeluropus lagopoides* (L.) *Thwaites* | 1 | 1 | 1 | 1 | 1 | 0.0001 | 0 | 0 | 0 | 0 | 0 | 75 |
| *Aerva javanica (Burm.f.) Juss. ex Schultes* | 0.0008 | 0.1391 | 1 | 1 | 0.1634 | 1 | 40.74 | 13.19 | 0 | 0 | 12.04 | 0 |
| *Aizoon canariense* L. | 0.0154 | 0.0052 | 1 | 1 | 1 | 1 | 21.43 | 28.57 | 0 | 0 | 0 | 0 |
| *Anastatica hierochuntica* L. | 1 | 1 | 1 | 1 | 0.0001 | 1 | 0 | 0 | 0 | 0 | 0 | 100 |
| *Arnebia hispidissima (Lehm.) DC* | 1 | 1 | 1 | 1 | 0.0001 | 1 | 0 | 0 | 0 | 0 | 87.5 | 0 |
| *Artemisia sieberi Besser* | 0.0001 | 1 | 1 | 1 | 1 | 1 | 75 | 0 | 0 | 0 | 0 | 0 |
| *Arthrocnemum macrostachyum (Moric.) K. Koch* | 1 | 1 | 1 | 1 | 1 | 0.0001 | 0 | 0 | 0 | 0 | 0 | 100 |
| *Avicennia marina (Forssk.) Vierh.* | 1 | 1 | 1 | 1 | 1 | 0.0004 | 0 | 0 | 0 | 0 | 0 | 50 |
| *Blepharis ciliaris* (L.) *B.L. Burtt.* | 0.0031 | 0.0269 | 1 | 1 | 1 | 0.2676 | 34.43 | 22.54 | 0 | 0 | 0 | 9.016 |
| *Capparis cartilaginea Decne.* | 1 | 0.0024 | 1 | 1 | 1 | 1 | 0 | 37.5 | 0 | 0 | 0 | 0 |
| *Capparis decidua (Forssk.) Edgew.* | 1 | 1 | 1 | 0.6142 | 1 | 0.001 | 0 | 0 | 0 | 1.786 | 0 | 42.86 |
| *Carthamus persicus* | 1 | 0.0004 | 1 | 1 | 1 | 1 | 0 | 50 | 0 | 0 | 0 | 0 |
| *Chrozophora oblongifolia (Delile) A.Juss. ex Spreng.* | 0.0312 | 1 | 1 | 1 | 1 | 0.0119 | 21.71 | 0 | 0 | 0 | 0 | 26.32 |
| *Citrullus colocynthis* (L.) *Schrad* | 0.0001 | 1 | 1 | 0.6574 | 1 | 0.1297 | 61.44 | 0 | 0 | 3.191 | 0 | 13.3 |
| *Cleome arabica* L. | 0.0001 | 1 | 1 | 1 | 1 | 1 | 62.5 | 0 | 0 | 0 | 0 | 0 |
| *Cleome droserifolia (Forssk.) Del.* | 1 | 1 | 1 | 0.0001 | 0.0006 | 1 | 0 | 0 | 0 | 55.26 | 44.74 | 0 |
| *Cocculus pendulus* | 1 | 0.0001 | 1 | 1 | 1 | 0.2732 | 0 | 56.25 | 0 | 0 | 0 | 6.25 |
| *Cyperus bulbosus Vahl* | 1 | 1 | 1 | 1 | 1 | 0.0001 | 0 | 0 | 0 | 0 | 0 | 100 |
| *Erodium oxyrhinchum M.Bieb.* | 1 | 1 | 1 | 1 | 1 | 0.1637 | 0 | 0 | 0 | 0 | 0 | 12.5 |
| *Fagonia indica Burm.f.* | 0.0012 | 0.0001 | 1 | 1 | 1 | 1 | 36.43 | 45 | 0 | 0 | 0 | 0 |
| *Ferula sinaica Boiss.* | 1 | 0.0001 | 1 | 1 | 1 | 1 | 0 | 75 | 0 | 0 | 0 | 0 |
| *Ficus cordata* ssp. *salicifolia (Vahl) C.C. Berg.* | 1 | 0.0028 | 1 | 1 | 1 | 1 | 0 | 37.5 | 0 | 0 | 0 | 0 |
| *Forsskaolea tenacissima* L. | 0.0037 | 0.0197 | 1 | 0.2169 | 1 | 1 | 33.46 | 24.04 | 0 | 10.58 | 0 | 0 |
| *Gomphocarpus sinaicus Boiss.* | 0.0001 | 1 | 1 | 1 | 1 | 1 | 87.5 | 0 | 0 | 0 | 0 | 0 |
| *Haloxylon salicornicum (Moq.) Bunge* | 0.0004 | 0.0044 | 0.4506 | 0.9983 | 1 | 0.9177 | 32.76 | 27.35 | 17.09 | 6.648 | 4.748 | 11.4 |
| *Heliotropium curassavicum* L. | 1 | 1 | 1 | 1 | 1 | 0.0001 | 0 | 0 | 0 | 0 | 0 | 75 |
| *Hyoscyamus muticus* L. | 0.0385 | 1 | 1 | 1 | 1 | 0.0038 | 18 | 0 | 0 | 0 | 0 | 32.5 |
| *Hyphaene thebaica* | 1 | 0.0514 | 1 | 1 | 0.095 | 0.0213 | 0 | 18.75 | 0 | 0 | 15.63 | 23.44 |
| *Iphiona scabra* | 1 | 0.0001 | 1 | 1 | 1 | 0.0754 | 0 | 68.48 | 0 | 0 | 0 | 16.3 |
| *Lasiurus scindicus Henrard* | 1 | 1 | 1 | 1 | 1 | 0.0001 | 0 | 0 | 0 | 0 | 0 | 100 |
| *Lavandula coronopifolia Poir* | 1 | 0.029 | 1 | 1 | 1 | 1 | 0 | 25 | 0 | 0 | 0 | 0 |
| *Lavandula pubescens Decne.* | 0.0219 | 0.0015 | 1 | 1 | 1 | 1 | 22.92 | 39.58 | 0 | 0 | 0 | 0 |
| *Leptadenia pyrotechnica (Forssk.) Decne.* | 1 | 0.0001 | 0.3064 | 0.1034 | 1 | 1 | 0 | 67.16 | 10.26 | 16.42 | 0 | 0 |
| *Leptochloa fusca* (L.) *Kunth.* | 0.0001 | 1 | 1 | 1 | 1 | 1 | 62.5 | 0 | 0 | 0 | 0 | 0 |
| *Lycium shawii Roem & Schult* | 0.0077 | 0.0001 | 0.9528 | 0.9973 | 1 | 0.5474 | 29.39 | 38.17 | 8.349 | 4.962 | 2.338 | 15.27 |
| *Maerua crassifolia Forssk.* | 1 | 1 | 1 | 1 | 1 | 0.0001 | 0 | 0 | 0 | 0 | 0 | 87.5 |
| *Medicago laciniata var. brachyacantha Boiss.* | 1 | 1 | 1 | 1 | 0.0007 | 1 | 0 | 0 | 0 | 0 | 50 | 0 |
| *Nitraria retusa (Forssk.) Asch* | 1 | 1 | 1 | 1 | 1 | 0.0001 | 0 | 0 | 0 | 0 | 0 | 100 |
| *Ochradenus baccatus Del.* | 0.0007 | 0.0001 | 0.5117 | 1 | 0.8879 | 0.9767 | 36.69 | 39.05 | 11.83 | 0 | 5.695 | 2.959 |

**Table 1.** *Cont.*

| Indicator Species (IS) | ELe1 | ELe2 | ELe3 | ELe4 | ELe5 | ELe6 | ELe1 % | ELe2 % | ELe3 % | ELe4 % | ELe5 % | ELe6 % |
|---|---|---|---|---|---|---|---|---|---|---|---|---|
| *Panicum turgidum* Forssk. | 1 | 0.0041 | 1 | 1 | 1 | 0.0001 | 0 | 31.4 | 0 | 0 | 0 | 50.87 |
| *Peganum harmala* L. | 0.0025 | 1 | 1 | 1 | 1 | 1 | 37.5 | 0 | 0 | 0 | 0 | 0 |
| *Pergularia tomentosa* L. | 0.0208 | 0.0195 | 1 | 0.2022 | 1 | 1 | 25.38 | 25.38 | 0 | 11.54 | 0 | 0 |
| *Phoenix caespitosa* | 1 | 0.0269 | 1 | 1 | 1 | 1 | 0 | 25 | 0 | 0 | 0 | 0 |
| *Phragmites australis* (Cav.) Trin. Ex. Steudel | 1 | 0.0009 | 1 | 1 | 1 | 0.1102 | 0 | 35.71 | 0 | 0 | 0 | 10.71 |
| *Plicosepalus acaciae* (Zucc.) Wiens & Polhill | 1 | 0.0044 | 1 | 1 | 1 | 0.0005 | 0 | 30.17 | 0 | 0 | 0 | 38.79 |
| *Pulicaria incisa* (Lam.) DC | 0.0104 | 0.229 | 1 | 0.1268 | 1 | 1 | 28.79 | 9.091 | 0 | 13.64 | 0 | 0 |
| *Pycnocycla saxatilis* Danin, Hedge & Lamond | 0.0033 | 1 | 1 | 1 | 1 | 1 | 37.5 | 0 | 0 | 0 | 0 | 0 |
| *Reseda muricata* C.Presl | 0.0034 | 1 | 1 | 1 | 1 | 1 | 37.5 | 0 | 0 | 0 | 0 | 0 |
| *Retama raetam* (Forssk.) Webb. | 0.0001 | 1 | 1 | 1 | 1 | 1 | 100 | 0 | 0 | 0 | 0 | 0 |
| *Rhazya stricta* Decne. | 0.0001 | 1 | 0.2617 | 0.5469 | 0.9601 | 0.0629 | 39.35 | 0 | 17.22 | 12.73 | 5.093 | 23.15 |
| *Salsola jordanicola* Eig | 0.0125 | 0.1611 | 1 | 1 | 1 | 1 | 30.56 | 9.722 | 0 | 0 | 0 | 0 |
| *Salvadora persica* L. | 1 | 1 | 1 | 1 | 1 | 0.0003 | 0 | 0 | 0 | 0 | 0 | 50 |
| *Seidlitzia rosmarinus* Bunge ex Boiss. | 1 | 1 | 1 | 1 | 1 | 0.0001 | 0 | 0 | 0 | 0 | 0 | 75 |
| *Senna holosericea* (Fresen) Greuter | 1 | 1 | 1 | 1 | 0.0001 | 1 | 0 | 0 | 0 | 0 | 100 | 0 |
| *Stipa capensis* Thunb. | 0.0001 | 1 | 1 | 1 | 1 | 1 | 87.5 | 0 | 0 | 0 | 0 | 0 |
| *Stipagrostis plumosa* (L.) Munro ex T.Anderson | 1 | 1 | 1 | 1 | 1 | 0.1616 | 0 | 0 | 0 | 0 | 0 | 12.5 |
| *Tamarix aphylla* (L.) Karst | 1 | 1 | 1 | 1 | 1 | 0.0035 | 0 | 0 | 0 | 0 | 0 | 37.5 |
| *Tamarix nilotica* (Ehrenb.) Bunge | 1 | 1 | 1 | 1 | 0.0004 | 0.0001 | 0 | 0 | 0 | 0 | 42.11 | 57.89 |
| *Tephrosia purpurea* ssp. *apollinea* (Del.) Hosni | 1 | 1 | 1 | 1 | 0.0001 | 1 | 0 | 0 | 0 | 0 | 100 | 0 |
| *Traganum nudatum* Delile | 1 | 1 | 1 | 1 | 1 | 0.004 | 0 | 0 | 0 | 0 | 0 | 37.5 |
| *Vachellia flava* (Forssk.) Kyal. & Boatwr | 1 | 1 | 1 | 1 | 1 | 0.0001 | 0 | 0 | 0 | 0 | 0 | 100 |
| *Vachellia gerrardii* (Benth.) | 0.0001 | 0.3625 | 1 | 1 | 1 | 1 | 75.35 | 5.208 | 0 | 0 | 0 | 0 |
| *Vachellia oeforta* (Forssk) Kyal. & Boatwr | 1 | 1 | 1 | 1 | 1 | 0.1637 | 0 | 0 | 0 | 0 | 0 | 12.5 |
| *Vachellia tortilis* (Forssk.) | 1 | 0.0001 | 0.4705 | 0.0683 | 0.2947 | 1 | 0 | 50.25 | 10.55 | 23.62 | 14.07 | 0 |
| *Vachellia tortilis* subsp. *raddiana* (Savi) Brenan | 1 | 0.0002 | 0.9445 | 0.4809 | 0.7695 | 0.0028 | 0 | 37.56 | 5.183 | 13.17 | 8.78 | 32.2 |
| *Vachellia tortilis* (Forssk.) | 1 | 1 | 1 | 1 | 1 | 0.0001 | 0 | 0 | 0 | 0 | 0 | 100 |
| *Zilla spinosa* (L.) Prantl | 0.0001 | 1 | 1 | 1 | 1 | 1 | 100 | 0 | 0 | 0 | 0 | 0 |
| *Ziziphus spina-christi* (L.) Desf. | 1 | 0.0001 | 1 | 1 | 0.6201 | 1 | 0 | 95.65 | 0 | 0 | 1.63 | 0 |
| *Zygophyllum coccineum* L. | 1 | 1 | 1 | 1 | 1 | 0.0001 | 0 | 0 | 0 | 0 | 0 | 100 |

Details of species domination expressed as the important value of species (IVs) are presented in Table 2. The important value of each species was based on the total relative plant cover and density at the studied elevations. For elevation 1 (above 1000 m a.s.l.), *Haloxylon salicornicum (Moq.) Bunge* was the dominant species (Ivs = 50.02), and the second most dominant species was *Retama raetam (Forssk.) Webb.* (Ivs = 36.29). Other important species included *Zilla Spinosa (L.) Prantl* (Ivs = 35.4), *Rhazya stricta Decne* (Ivs = 29.57), *Vachellia gerrardii (Benth.)* (Ivs = 19.64), and *Lycium shawii Roem & Schult* (Ivs = 19.09) (Table 1). For elevation 2 at 800 m (a.s.l.), *Haloxylon salicornicum (Moq.) Bunge* was the dominant species (Ivs = 44.24), and the co-dominant species *Vachellia tortilis* subsp. *Raddiana (Savi) Brenan* recorded Ivs = 27.49, while the other four species *Vachellia tortilis (Forssk.), Ochradenus baccatus Del., Leptadenia pyrotechnica (Forssk.) Decne., Lycium shawii Roem & Schult* achieved important values (Ivs = 26.41, 23.51, 23.03, 20.83, respectively). For elevation 3 (600 m a.s.l.), *Haloxylon salicornicum (Moq.) Bunge* was the most dominant species (Ivs = 95.06), followed by the co-dominant species *Rhazya stricta Decne.* (Ivs = 64.64), while the other important species were *Vachellia tortilis (Forssk.)* (Ivs = 37.23), *Leptadenia pyrotechnica (Forssk.) Decne.* (Ivs = 33.03), *Lycium shawii Roem & Schult* (Ivs = 27.86), *Vachellia tortilis* subsp. *Raddiana (Savi) Brenan* (Ivs = 23.40), and *Ochradenus baccatus Del.* (Ivs = 18.74). Similarly, the most dominant species at elevation 4 (400 m a.s.l.) was *Haloxylon salicornicum (Moq.) Bunge* (IV = 56.52), followed by the second-most dominant species *Rhazya stricta Decne.* (IV = 52.56), and five species, *Vachellia tortilis (Forssk.), Vachellia tortilis* subsp. *Raddiana (Savi) Brenan, Cleome droserifolia (Forssk.) Del., Leptadenia pyrotechnica (Forssk.) Decne.,* and *Lycium shawii Roem & Schult*, recorded important values (49.86, 38.83, 27.64, 25.72, 14.47, respectively). For elevation 5 (200 m a.s.l.), *Haloxylon salicornicum (Moq.) Bunge* (Ivs = 37.76) was the dominant species, and the second dominant species was *Vachellia tortilis (Forssk.)* (Ivs = 34.72); the other important species were *Tamarix nilotica (Ehrenb.) Bunge, Rhazya stricta Decne., Vachellia tortilis* subsp. *Raddiana (Savi) Brenan, Senna holosericea (Fresen) Greuter, Tephrosia purpurea* ssp. *Apollinea (Del.) Hosni, Cleome droserifolia (Forssk.) Del.,* and *Hyphaene thebaica* (L.)

*Mart.* (33.68, 30.95, 29.47, 25.33, 21.13, 18.40, 14.63, respectively). At elevation 6 (less than 100 m a.s.l.), the dominant species was *Vachellia tortilis (Forssk.)* (Ivs = 32.62), while the co-dominant species (Ivs = 25.45) was *Haloxylon salicornicum (Moq.) Bunge.* Meanwhile, four species: *Zygophyllum coccineum L.*, *Rhazya stricta Decne.*, *Vachellia tortilis* subsp. *Raddiana (Savi) Brenan*, and *Tamarix nilotica (Ehrenb.) Bunge.* Exhibited an important value index (23.44, 19.44, 16.79, 14.51, respectively).

**Table 2.** The important value of species domination (IVs) at the studied elevations, the most dominant, co-dominance, and important species. Values are the average of the important values of species based on the relative cover and density.

| Elevation Index | 1st Dominant | 2nd Dominant | Important Species |
|---|---|---|---|
| *Elevation 1 ≥ 1000 m (a.s.l)* | *Haloxylon salicornicum (Moq.) Bunge [50.02]* | *Retama raetam (Forssk.) Webb. [36.29]* | *Zilla Spinosa* (L.) *Prantl [35.47]* |
| | | | *Rhazya stricta Decne. [29.57]* |
| | | | *Vachellia gerrardii (Benth.) [19.64]* |
| | | | *Lycium shawii Roem & Schult [19.09]* |
| *Elevation 2 = 800 m (a.s.l)* | *Haloxylon salicornicum (Moq.) Bunge [44.24]* | *Vachellia tortilis* subsp. *raddiana (Savi) Brenan [27.49]* | *Vachellia tortilis (Forssk.) [26.41]* |
| | | | *Ochradenus baccatus Del. [23.51]* |
| | | | *Leptadenia pyrotechnica (Forssk.) Decne. [23.03]* |
| | | | *Lycium shawii Roem & Schult [20.83]* |
| *Elevation 3 = 600 m (a.s.l.)* | *Haloxylon salicornicum (Moq.) Bunge [95.06]* | *Rhazya stricta Decne. [64.64]* | *Vachellia tortilis (Forssk.) [37.23]* |
| | | | *Leptadenia pyrotechnica (Forssk.) Decne. [33.03]* |
| | | | *Lycium shawii Roem & Schult [27.86]* |
| | | | *Vachellia tortilis* subsp. *raddiana (Savi) Brenan [23.40]* |
| | | | *Ochradenus baccatus Del. [18.74 ]* |
| *Elevation 4 = 400 m (a.s.l.)* | *Haloxylon salicornicum (Moq.) Bunge [56.52]* | *Rhazya stricta Decne. [52.56]* | *Vachellia tortilis (Forssk.) [49.86]* |
| | | | *Vachellia tortilis* subsp. *raddiana (Savi) Brenan [38.83]* |
| | | | *Cleome droserifolia (Forssk.) Del. [27.64]* |
| | | | *Leptadenia pyrotechnica (Forssk.) Decne. [25.72]* |
| | | | *Lycium shawii Roem & Schult [14.47]* |
| *Elevation 5 = 200 m (a.s.l.)* | *Haloxylon salicornicum (Moq.) Bunge [37.76]* | *Vachellia tortilis (Forssk.) [34.72]* | *Tamarix nilotica (Ehrenb.) Bunge [33.68]* |
| | | | *Rhazya stricta Decne. [30.95]* |
| | | | *Vachellia tortilis* subsp. *raddiana (Savi) Brenan [29.47]* |
| | | | *Senna holosericea (Fresen) Greuter [25.33]* |
| | | | *Tephrosia purpurea* ssp. *apollinea (Del.) Hosni [21.13] [18.40]* |
| | | | *Cleome droserifolia (Forssk.) Del.* |
| | | | *Hyphaene thebaica* (L.) *Mart. [14.63]* |
| *Elevation 6 ≤ 100 m (a.s.l.)* | *Vachellia tortilis (Forssk.) [32.62]* | *Haloxylon salicornicum (Moq.) Bunge [25.45]* | *Zygophyllum coccineum* L. *[23.44]* |
| | | | *Rhazya stricta Decne. [19.44]* |
| | | | *Vachellia tortilis* subsp. *raddiana (Savi) Brenan [16.79]* |
| | | | *Tamarix nilotica (Ehrenb.) Bunge [14.51]* |

Vegetation–soil analysis

The soil profile of 15 parameters showed significant variations in all measured elements; however, pH and OM% were not significant among the elevational zones (Table 3). The moisture content MC% and sand percentage were significantly prominent at elevation 1 ($\geq$1000 m) and elevation 2 (800 m), respectively. Elevation 3 (600 m) was associated with high potassium, bicarbonate, calcium carbonate, clay, and silt content. Elevation 4 (400 m) and elevation 5 (200 m) were characterized by high percentages of sand and silt, respectively. While elevation 6 ($\leq$100 m a.s.l.) achieved the highest average of EC, Na, Mg, Ca, Cl, $SO_4$, silt, and MC (Table 3).

**Table 3.** Soil chemical and physical properties of the studied elevations. Values are mean $\pm$ standard errors. "a" is a letter within each row showed significant variation at $p < 0.05$ (Duncan's test). (ns) denotes not significant, * $p < 0.05$, ** $p < 0.01$, *** $p < 0.001$ at the degree of freedom (*df*) for the elevations (n − 1) = 5 and replications (n − 1) = 7.

| Parameters | Altitudes | | | | | | MS | F | p Value |
|---|---|---|---|---|---|---|---|---|---|
| | Alt1 | Alt2 | Alt3 | Alt4 | Alt5 | Alt6 | | | |
| pH | 8.167 ±0.04 | 8.16 ±0.084 | 7.937 ±0.07 | 8.027 ±0.103 | 8.18 ±0.068 | 8.085 ±0.084 | 0.073 | 1.43 | 0.236 ns |
| EC (dS.m$^{-1}$) | 0.38 ±0.033 | 0.301 ±0.027 | 1.613 ±0.598 | 0.672 ±0.186 | 2.543 ±1.566 | **6.531** **±3.096 a** | 45.188 | 2.74 | 0.034 * |
| TDS | 243.25 ±21.191 | 189.24 ±17.941 | 1063.2 ±378.96 | 444.58 ±120.56 | **1681** **±994.08** | 4227.3 ±1968.9 | 1.896^07 | 2.83 | 0.030 * |
| K (meq/L) | 0.343 ±0.048 | 0.407 ±0.08 | **2.707** **±0.989 a** | 0.936 ±0.13 | 0.991 ±0.18 | 0.837 ±0.227 | 5.951 | 3.99 | 0.005 ** |
| Na (meq/L) | 0.982 ±0.307 | 0.613 ±0.145 | 6.818 ±4.015 | 1.147 ±0.381 | 22.808 ±13.544 | **39.586** **±20.028 a** | 2037.53 | 2.5 | 0.048 * |
| Ca (meq/L) | 2.537 ±0.287 | 2.68 ±0.352 | 9.06 ±1.403 | 3.976 ±0.895 | 30.57 ±22.228 | **59.729** **±20.366 a** | 4239.4 | 3.24 | 0.016 * |
| Mg (meq/L) | 1.181 ±0.166 | 0.95 ±0.097 | 3.693 ±0.865 | 2.591 ±0.932 | 6.063 ±4.057 | **82.906** **±39.957 a** | 8563.5 | 3.98 | 0.005 ** |
| HCO$_3$ (%) | 1.62 ±0.195 | 1.411 ±0.153 | **4.127** **±1.031 a** | 1.873 ±0.324 | 1.256 ±0.164 | 1.041 ±0.125 | 10.289 | 5.85 | **0.0005** **\*\*\*** |
| Cl (meq/L) | 0.796 ±0.16 | 1.337 ±0.141 | 6.271 ±2.657 | 2.098 ±0.678 | 54.956 ±43.825 | **176.32** **±91.981 a** | 39061.7 | 2.73 | 0.035 * |
| SO$_4$ (meq/L) | 2.633 ±0.483 | 1.845 ±0.426 | 12.451 ±3.678 | 4.506 ±1.381 | 27.512 ±18.981 | **128.19** **±54.981 a** | 19434.5 | 4.18 | 0.004 ** |
| CaCO$_3$ % | 9.117 ±1.923 | 2.872 ±0.528 | **17.044** **±1.733 a** | 6.052± 1.258 | 4.267 ±0.69 | 5.962 ±1.041 | 208.008 | 17.09 | **<0.0001** **\*\*\*** |
| O.M % | 0.257 ±0.051 | 0.392 ±0.141 | 0.471 ±0.11 | 0.171 ±0.08 | 0.197 ±0.038 | 0.278 ±0.046 | 0.107 | 1.6 | 0.185 ns |
| Clay % | 5.637 ±0.843 | 3.3 ±0.50 | **7.023** **±0.628 a** | 4.018 ±0.727 | 6.425 ±0.413 | 3.362 ±0.692 | 20.892 | 6.29 | **0.0003** **\*\*\*** |
| Silt % | 10.836 ±2.185 | 2.775 ±0.936 | **12.361** **±1.969 a** | 4.835 ±1.565 | **11.726** **±1.092** | **12.256** **±3.566** | 141.91 | 3.59 | 0.010 * |
| Sand % | 83.055 ±2.999 | 93.948 ±1.002 | 78.86 ±2.6056 | **88.95** **±2.895** | 79.844 ±1.758 | 82.76 ±4.147 | 268.21 | 4.54 | 0.002 ** |
| MC % | **0.638** **±0.112** | 0.546 ±0.199 | 0.2013 ±0.043 | 0.11 ±0.042 | 0.202 ±0.031 | 0.71 ±0.215 | 0.539 | 4.05 | 0.005 ** |

Canonical component analysis (CCA) showed that elevation 1 and elevation 2 shared a correlation between sand content and the important species *V. gerrardii*, *Z. spinosa*, *R. raetam*, *O. baccatus*, and *L. shawii* on the bottom-left side of the CCA biplot. For the left upper CCA biplot, a positive correlation was found between *H. salcornicum*, *R. stricta*, *L. pyrotechnica*, and *V. raddiana* and the five parameters K, HCO$_3$, CaCO$_3$, organic matter OM, and clay content at elevations 3 and 4. Finally, elevations 5 and 6 showed close trends at the right upper CCA biplot; *S. holosericea*, *T. apollinea*, *T. nilotica*, *H. thebaica*, and *Z. coccineum* were significantly correlated with silt content, salinity, Ca, Mg, Na, Cl, and sulfate (Figure 6).

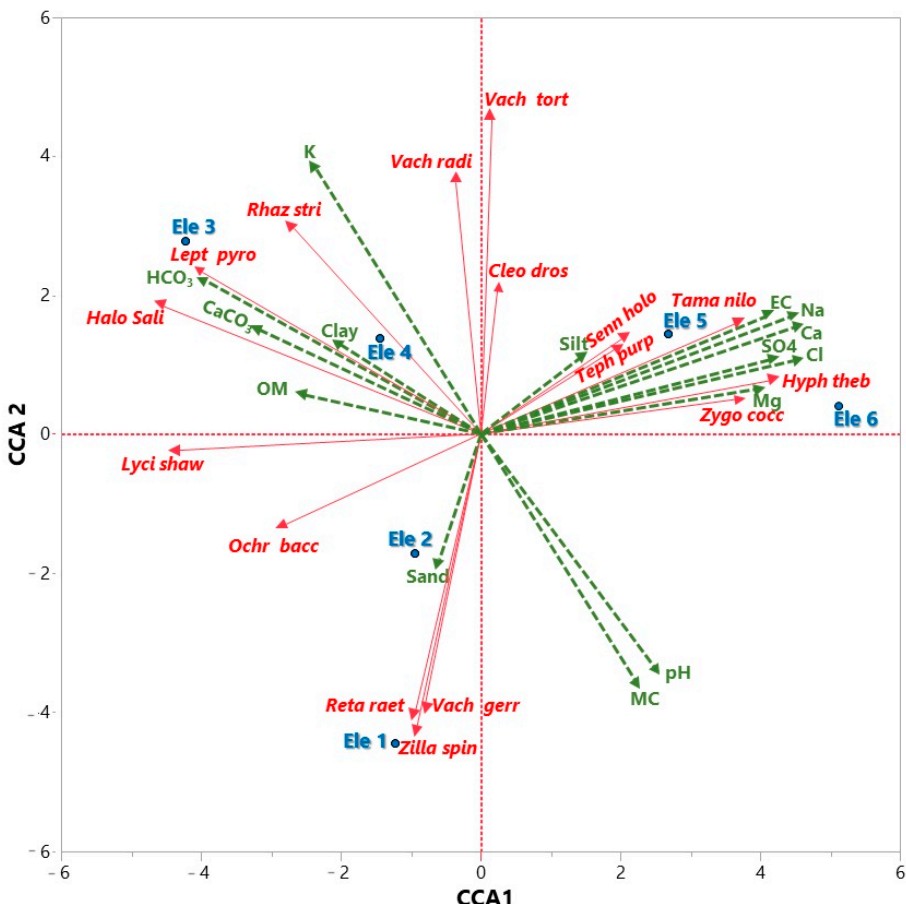

**Figure 6.** Canonical correspondence analysis (CCA) shows the correlation between the soil variables and the dominant and important species representing the studied elevations. Cleo dros: *Cleome droserifolia*; Halo Sali: *Haloxylon salicornicum*; Hyph theb: *Hyphaene thebaica*; Lept pyro: *Leptadenia pyrotechnica*; Lyci shaw: *Lycium shawii*; Ochr back: *Ochradenus baccatus*; Reta reat: *Retama raetam*; Rhaz stri: *Rhazya stricta*; Senn holo: *Senna holosericea*; Tama nilo: *Tamarix nilotica*; Teph purp: *Tephrosia purpurea*; Vach gerr: *Vachellia gerrardii*; Vach tort: *Vachellia tortilis*; Vach radd: *Vachellia raddiana*; Zill spin: *Zilla spinosa*; Zygo cocc: *Zygophyllum coccineum*. OM: organic matter; EC: electrical conductivity; MC: moisture content; K: potassium; Ca: calcium; Mg: magnesium; Na: sodium; $SO_4$: sulfate; Cl: chlorine; $CaCO_3$: calcium carbonate; $HCO_3$: bicarbonate; and pH: potential hydrogen.

A heatmap illustrated Pearson's correlation between dominant, co-dominant, and important species and soil variables (Figure 7). *Haloxylon salicornicum* and *R. stricta* showed a positive correlation with K, $HCO_3$, $CACO_3$, and clay content. In contrast, they showed a negative correlation with pH and MC, while other soil variables were not significant. *Vachilla tortilis* and *V. raddiana* had negative correlations with MC. *Leptadenia pyrotechnica*, *L. shawii*, and *O. baccatus* were positively correlated with OM, $CaCO_3$, and $HCO_3$, but negatively correlated with pH, Na, Ca, and Mg. *Cleome droserifolia* revealed a negative correlation with OM and MC. *Zilla spinosa*, *V. gerrardii*, and *R. raetam* had a positive correlation with MC. There was a positive correlation between measured parameters pH and salinity and most cations Na, Ca, Mg, and important species *H. thebaica* and *T. nilotica*. For other important species, *S. holosericea* and *T. apollinea* showed positive correlations with pH, Na, Ca, clay, and silt. As an important species in elevation 6 ($\leq$100 m a.s.l.), *Zygophyllum coccineum* was positively correlated with most soil variables, including salinity, cations, anions, and MC.

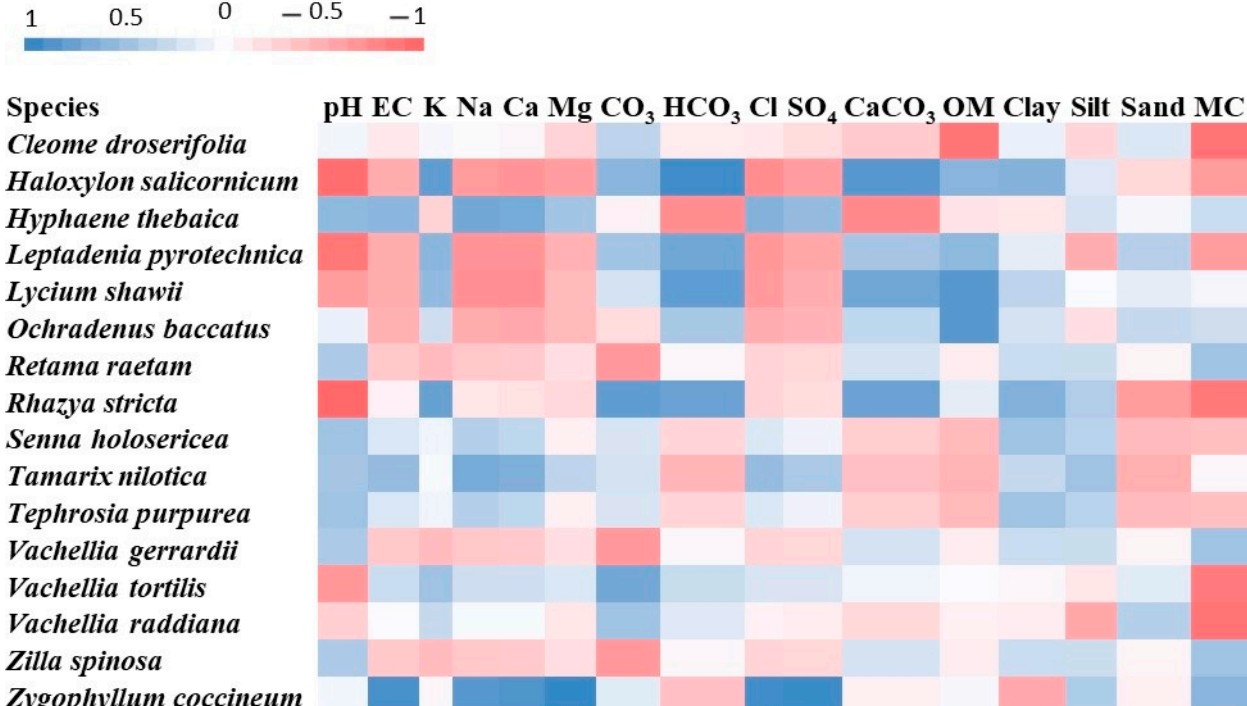

**Figure 7.** Pearson's correlation heatmap between the soil variables and the dominant, co-dominant, and important associated plant species within the studied elevations. EC: electrical conductivity; OM: organic matter; MC: moisture content; K: potassium; Ca: calcium; Mg: magnesium; Na: Sodium; $SO_4$: sulfate; Cl: chlorine; $CaCO_3$: Calcium carbonate; $HCO_3$: bicarbonate; and pH: potential hydrogen.

## 4. Discussion

A comprehensive survey of diversity traits for all elevations involving various habitats highlighted 70 species belonging to 33 families, of which Fabaceae, Poaceae, Asteraceae, and Chenopodiaceae were the most ubiquitous families, with distinctive efficiency of prevalence, enabling their seeds to be dispersed in xeric environments [46]. The studied species were also categorized according to chorological affinities, as 47% of species were deemed monoregional, whereas 11 species were identified as Saharo-Arabian (SA), and 10 as Sudanian (SUD). It was determined that 35.7% were 16 ioregional, whereas 13 species comprised the Saharo–Sindian and Sudano–Zambezian group (SS + SZ). The relatively high contribution of these phytogeographical groups reflects the general arid climate pattern of the Arabian Peninsula [47]. The life form of identified species, denoted as chamaephytes and phanerophytes, and growth habits such as herbaceous and shrub forms, dominated all surveyed elevations; these patterns are very common in arid environments, especially in the north-west region of the Arabian Peninsula, in agreement with other studies [48,49].

The present work Investigated the diversity indices observed with elevational variation in PMBSNR. Alpha diversity showed that elevation ≥ 1000 m a.s.l. (El1) and elevation 800 a.s.l. (El2), and elevation ≤ 100 m a.s.l. (El6) recorded high taxa-species richness, Shannon, and Simpson values; this may be attributed to variation in climate features according to our metrology data, i.e., the humidity and temperature were significantly different at these elevational gradients compared to other elevations. Elevations 300 m and 400 m a.s.l. (El3, El4) had low alpha diversity and high evenness values. Analysis of the Bray–Curtis cluster and NMDS showed a likelihood of a resemblance in the environmental conditions of the studied sites; in these habitats, soil analysis (CCA) demonstrated the richness of $HCO_3$, $CaCO_3$, and K, which could be associated with similar plant species that flourished with a high percentage of these chemical elements at elevations 3 and 4. Beta diversity (Whittaker index) showed that elevation ≥ 1000 m a.s.l. (El1) and elevation 800 a.s.l. (El2)

had the highest species turnover; respective species colonized at a significant distance from the coastline. This suggests that the unique climate variables (lower relative humidity and temperatures) at these elevations have notably influenced the plant species that can thrive in such conditions. These elevations are far from the coastline, so plant species adapted accordingly, exhibiting low salt tolerance and more xerophytes [18,50].

The current study provides insight into indicator species, important values of plant communities, and relative soil in terms of elevational variation. The highest contribution of indicator species (six plant species) recorded at elevation $\leq$ 100 m (a.s.l.) (El6) may be the result of increased environmental heterogeneity of micro-habitats represented in this range compared to other elevations, which had only one or two species [51]. The elevational gradient is often a proxy of climate variation on a broad spatial scale [52]. It was found in the present study that elevation $\leq$ 100 m (a.s.l.) (El6) had the highest relative humidity, incentivizing the development of several indicator species and supporting the co-occurrence of species typical of contrasting habitat types, even within a couple of meters [51]. Therefore, changes in habitat patches and species interactions across PMBSNR can affect the distribution and abundance of indicator species [53].

Four species were widespread and overlapping for plant communities at all elevational gradients: *Haloxylon salicornicum*, *Vachellia tortilis* subsp. *Raddiana*, *Rhazya stricta*, and *Vachellia tortilis*. These species are unique to desert lands in the Arabian Peninsula [47,54]; it is worth explaining that environmental conditions, particularly soil characteristics, can significantly affect plant communities' colonization efficiency. CCA appears to be more pronounced; accordingly, both *H. salicorncum* and *R. stricta* in Elevation 600 (EL3) may be tolerant of soils with high $CO_3$, $HCO_3$, and $CaCO_3$. Moreover, the competitive ability, a merit of these two species, often plays a role in their prevalence, promoting domination in most studied elevations [55].

One key factor is the dominant and co-dominant species' ability to tolerate a broad range of environmental conditions. For example, some plants and animals can thrive in both wet and dry environments or in hot and cold climates. This flexibility may be attributed to their phenotypic plasticity, which is linked to their functional diversity in terms of nutritional strategies [56]. Therefore, super-dominant species among elevations, such as *Haloxylon salicornicum*, can occupy diverse habitats and exploit different resources, increasing their chances of survival at the most elevated gradients [57].

Notably, elevation 3, at which alpha and beta diversity were very low and no species indicator was found, may reflect the increment in environmental stress compared to other elevations. In other words, only species presented by vegetation communities as dominant and co-dominant species may survive in harsh, low-nutrient soil. Other important species colonized at either elevation $\leq$ 100 m or elevation 200 m (a.s.l.), where the vegetation zone meets the Red Sea shore and is influenced by the coastal atmosphere, exhibited positive correlations with EC, Na, $SO_4$, Cl, Ca, and Mg. Most of these species are classified as salt-tolerant plants (halophytes), which grow in saline environments, in line with the results obtained by [58,59]. Other important species, such as *Retama retam*, *Vachellia gerrardii*, *Zilla spinosa*, *Lycium shwaii*, and *Leptadenia pyrotechnica*, were correlated with sandy soil as prominent soil properties in elevation $\geq$ 1000 m and elevation 800 m (a.s.l.). According to field observations at elevations above 1000 m and 800 m a.s.l., the topography contains a high sand percentage, appearing as patches among rugged rocks; thus, these plant species' deep root systems have adapted to grow and develop in these elevational conditions [50].

This study found that PMBSNR had two distinguished microclimates, according to the data obtained from two weather stations: one in the uplands (Shigry station, 1000 m a.s.l.). and one on the coastline (Almuwaylih station, 100 m a.s.l.). Phytogeography of sites at 1000 m a.s.l. belonging to the Saharo-Arabian region as a general pattern of the Arabian Peninsula, which is characterized by high aridity as a part of interior climate zonation, identified species at elevation ($\leq$100 m a.s.l.) would represent the Sudanese region's climate, with high temperatures and high humidity throughout the year [27,47]. These variations in microclimates at different elevations lead to disparities in plant diversity metrics (alpha

diversity indices, indicator species, and soil characteristics). To clarify, the highest species richness was the obvious diversity index for sites at these comparable elevations, as well as the indicator species (Table 1) *Retama raetam* and *Zilla spinosa*, which had distinctive microclimates as they grew on sand soil at elevations above 1000 m; correspondingly, eight indicators were identified as belonging to saline-tolerant species that flourish in high-salinity soil as a distinctive microclimate, and these existed at elevations beginning at 100 m from the seashore. The meteorology data were limited for other elevations (800 m, 600 m, 400 m, 200 m), at which there was no mini-weather station that could help interpret the distribution of the remaining species. However, the soil profile of these elevations can be an indicator of the effect of the local climate on vegetation distribution.

## 5. Conclusions

PMBSNR embraces distinguished climate ecosystems, particularly elevations $\geq$ 1000 m and $\leq$100 m (a.s.l.), which can create heterogeneity in microhabitats, exhibiting floristic diversity, distinct indicator species, and species domination in plant communities. Ongoing environmental changes, such as variability in soil characteristics, can manipulate plant functional traits, affecting species assemblage across interzonal elevations. Understanding ambient environmental factors is essential when determining ways to manage the xeric ecosystem. This study focused on a field survey and established microsite references from high elevations to coastline zones through phytosociology and soil profiles. Future work could investigate the long-term effects of microclimates at varying elevations. Replanting native species next to parent plants in similar populations and microhabitats using a habitat restoration protocol, such as the soil–seed bank technique, and manipulating seeds and seedlings through a symmetric plot design at each elevation is a possible strategy for creating a sustainable ecosystem. Monitoring microsites is necessary to explore vegetation dynamics and species distribution patterns and to determine how they respond to and survive the heterogeneity of the local climate. Such research could help us identify specific strategies and patterns of restoration that relate to successful management and conservation in PMBSNR.

**Supplementary Materials:** The following supporting information can be downloaded at: https://www.mdpi.com/article/10.3390/d15101081/s1, Table S1: Floristic analysis of the studied region at different elevations, Table S2: Species domination (IVs) of the studied region at different elevations, values are the average of the important values of species based on the relative cover and density.

**Funding:** Deanship of Scientific Research at University of Tabuk, project No. S-0271-1443.

**Institutional Review Board Statement:** Not applicable.

**Informed Consent Statement:** Not applicable.

**Data Availability Statement:** Not applicable.

**Acknowledgments:** The author appreciates the Deanship of Scientific Research at the University of Tabuk for supporting this work through funding No. S-0271-1443.

**Conflicts of Interest:** The author declares no conflict of interest.

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
