# Peer review of "Diversity of Indicator and Dominant Plant Species along Elevation Gradients in Prince Mohammad Bin Salman Nature Reserve, KSA"

_diversity, doi:10.3390/d15101081_

Round 1
Reviewer 1 Report
Elevation is a complex environmental gradient, associated globally with declining atmospheric pressure and temperature, whereas abiotic factors such as solar radiation or precipitation show more regional patterns. Across elevation gradients, changes in abiotic and biotic factors can therefore lead to strongly divergent selection, which may result in local adaptation. Ecologists have studied patterns of species richness along elevational gradients for a long time. For determining patterns of species richness, factors like climate, geographical area, and habitat heterogeneity were taken into account. Investigations on macroecological patterns have led to a general theory of species diversity and distributions and also to understanding of various local and regional environmental factors that drive them. Within this context, the present study recorded 70 species in 33 families, and elucidated floristic traits correlated with elevation in Crown Prince Mohammed Bin Salman Nature Reserve. The results show that microclimate, including temperature and relative humidity variations, was found to be a significant driver in the ecosystem, resulting in variation in plant diversity and species distribution at different elevations.
I think this study is a progress by deepening and expanding knowledge about plant elevation adaptation with a focus in Prince Mohammad bin Salman Nature Reserve, KSA. This manuscript highlights the need of more research to be investigated along elevation gradient for determining general pattern of species variation. Generally, some revision suggestions are listed below:
(1) Elevation is a complex environmental gradient, associated globally with declining atmospheric pressure and temperature, whereas abiotic factors such as solar radiation or precipitation show more regional patterns. It's better to add the above related content in Introduction. (e.g. Assessing Elevation-Based Forest Dynamics over Space and Time toward REDD+ MRV in Upland Myanmar, Remote Sensing 2022; Changes in Bird Community Structure on Mount Cameroon Driven by Elevational and Vertical Gradients, Diversity 2023; The Effects of Tropical Elevations and Associated Habitat Changes on Firefly (Coleoptera: Lampyridae) Diversity in Malaysia, Diversity 2023 ...)
(2) It's better to discuss the implications from plant species along elevation gradients in Prince Mohammad bin Salman Nature Reserve from the perspective of climate change.
(3) Please suggest the future research directions.
Reviewer 2 Report
The manuscript, titled "Exploring Plant Diversity and Dominance Along Elevation Gradients in Prince Mohammad bin Salman Nature Reserve, KSA," sheds light on previously unexplored variations in flora and vegetation distribution across the expansive, arid landscape of the Crown Prince Mohammed Bin Salman Nature Reserve. The study identified and documented 70 species from 33 plant families, unraveling the floristic traits associated with different elevations. The author conducted thorough vegetation analyses, identified indicator species based on their relative abundance, and elucidated patterns of species distribution in relation to elevation. Additionally, the author established that soil characteristics, such as physical structure and chemistry, significantly influenced species dominance. The study underscores the importance of considering microclimate effects when planning vegetation restoration or afforestation within this vital nature reserve.
I commend the author for their valuable contribution. The manuscript is well-crafted and has the potential to captivate a wide audience interested in invasive biology.
However, to enhance the manuscript's quality, I would like to provide the following feedback on various sections:
Title: Appropriate
Abstract:
The abstract lacks information about the methodology used for data collection. I recommend adding a brief description of the data collection methods.
The conclusion in the abstract could benefit from some expansion.
Introduction:
The introduction incorporates relevant literature; however, I suggest including more recent papers that are pertinent to the topic.
In the first sentence of the Introduction, I propose replacing "anthropological" with "anthropogenic" for clarity.
Material and Methods:
The methodology is well-documented, with clear and well-applied methods and robust statistical analyses. Please refer to the comments on Figure 1 in the Figures section.
Results:
The results are effectively presented, with graphical figures aiding in comprehension. The statistical methods used are sound.
Change "Zyophllaceous" to "Zygophyllaceae" to accurately represent the family.
Discussion:
The discussion offers valuable insights, with pertinent commentary and comparisons. I would like to emphasize that most of the references cited in this section are more than five years old.
Conclusion:
The conclusion is concise and to the point.
Figures:
Figure 1: The map requires refinement, including the addition of a title, scale, a north arrow, and the omission of the 1200m zone on the map.
Figure 3: The letters used in the figure need descriptions. For example, explain the significance of "SS" and "SZ."
Tables:
The tables provided are suitable.
References:
Many of the references cited in the manuscript are more than five years old, with a total of 50 out of 63 references falling into this category. Consider incorporating more recent sources to bolster the literature review.
Overall, this manuscript holds promise and is a valuable contribution to the field. Addressing the recommended revisions will further enhance its quality and appeal to a wider audience.
Reviewer 3 Report
Dear Dhafer,
Good effort! I believe the data you've gathered holds promise for potential publication in a journal. It's encouraging to see research originating from this region. However, there are some areas in the article that require improvement, particularly in streamlining the core message, refining the methodology, and presenting a coherent narrative. I encountered issues with the methodology; several components, particularly those related to cluster analysis and statistical testing, were missing, rendering the study non-reproducible. Additionally, the selection of traits lacked clarity and ecological justification, necessitating alignment with the study's specific constraints. While I've highlighted a few points for consideration, I refrained from providing an exhaustive list due to the absence of line numbers, which made the review process challenging.
Abstract
· Preservation and conservation are completely different research fields. To preserve an area is not to conserve, likewise, to conserve an area does not that it needs to be preserved. Please completely rethink the first sentence.
· I found that the data was well suited to a classical vegetation descriptive study (and the data are robust to support that), defining vegetation types, their ecological drivers. Then for me the secondary angle should be about how this type of work can support conservation.
Introduction
· Paragraph 1, sentence three. The spatial distribution of a species does not affect species incidence and abundance – rather, the distribution of species across a landscape, their positions relative to each other, provide us with insights about the role of abiotic and biotic processes that may affect their fitness.
Method
· Well balanced design, six elevation zones, eight sites each zone.
· 2.4 data analysis/
o I don’t know about the choice of traits. It appears that the author has thrown down lots of words (life-form, growth form, chorotype) in essence different schemes developed for different environments – why should we care about the selection you have made – how are they ecologically relevant to promoting the persistence of plants under the environmental conditions / ecological constraints presented?
Results
· What statistical testing was used for species richness analysis is not clear?
· Cluster analyses presented in the results but not described in the methods (NMDS is not a cluster analysis technique – so what has been used?), tools, software etc all missing)
· I can see that there are some very nice clusters (vegetation types) stratified along the environmental gradient). An ordered synoptic table would help.
· Figure 3. Chlorotype abbreviations not described.
Please run it through grammarly -- focus on clarity. There are many words that can be scrapped. Check also use of en type dash, which should be used from a to b (i.e., a–b). Instead the author has used the em-dash consistently (and incorrectly).
